# The Effect of Sex-Specific Differences on IL-10^−/−^ Mouse Colitis Phenotype and Microbiota

**DOI:** 10.3390/ijms241210364

**Published:** 2023-06-20

**Authors:** Maite Casado-Bedmar, Maryline Roy, Emilie Viennois

**Affiliations:** INSERM, U1149, Center of Research on Inflammation, Université de Paris, 75018 Paris, France; maite.casado@inserm.fr (M.C.-B.); maryline.roy@inserm.fr (M.R.)

**Keywords:** IBD, microbiota, colitis, Interleukin-10, mouse model, IL-10 knockout mice, female, sex differences

## Abstract

Sexual dimorphism is an important factor in understanding various diseases, including inflammatory bowel disease (IBD). While females typically exhibit stronger immune responses, the role of sex in IBD remains unclear. This study aimed to explore the sex-dependent differences and inflammatory susceptibility in the most extensively used IBD mouse model as they developed colitis. We monitored IL10-deficient mice (IL-10^−/−^) up to 17 weeks of age and characterized their colonic and fecal inflammatory phenotype, as well as their microbiota changes. Here, we originally identified IL-10^−/−^ female mice as more prone to developing intestinal inflammation, with an increase in fecal miR-21, and dysbiosis with more detrimental characteristics compared to males. Our findings provide valuable insights into the sex-based differences in the pathophysiology of colitis and emphasize the importance of considering sex in experimental designs. Moreover, this study paves the way for future investigations aiming at addressing sex-related differences for the development of adequate disease models and therapeutic strategies, ideally enabling personalized medicine.

## 1. Introduction

The importance of sex in understanding various diseases has become increasingly examined. Sex plays a role in modulating physiological processes, both in normal and pathological conditions [1]. Scientific evidence supports the existence of notable differences between males and females in their innate and adaptive immune response to antigens. Females exhibit stronger immune responses to infections and more robust antibody responses to vaccinations [2]. While this heightened immunoreactivity enables females to eliminate pathogens more efficiently than males, it also renders them more vulnerable to autoimmune and inflammatory diseases. Generally, females account for approximately 80% of all patients affected by autoimmunity [3]. 

Sex-based disparities in immune function could partially be explained by differential inflammatory cytokine production, secretion, and/or action between sex. Interleukin 10 (IL10) is considered one of the most important immunoregulatory cytokines, playing an essential role in the control and resolution of pro-inflammatory responses. Particularly in the gastrointestinal tract, IL10 is closely associated with the development of colitis [4]. Some studies have shown sex differences in immune cell responsiveness ex vivo [5]. For instance, the secretion of IL10 by male peripheral mononuclear cells (PBMCs) after activation with lipopolysaccharide (LPS) is significantly lower compared to females [5]. In other situations, sex-based differences are not explained by differences in cytokine concentration. Interestingly, recent studies have shown male immune cells have greater responsiveness to the anti-inflammatory effects of IL10 [6]. To evaluate the in vivo role of IL10, a large amount of research has been carried out in IL10-deficient mice. IL10 knockout mice (IL-10^−/−^) are frequently used to investigate immunoregulatory pathways. More importantly, these animals are known to develop spontaneous enterocolitis characterized by extensive mucosal hyperplasia, crypt abscesses, ulcers, mucin depletion, and intestinal wall thickening [7]. Thus, IL-10^−/−^ mice are the most common animal model used to elucidate the pathogenesis of human gastrointestinal diseases such as inflammatory bowel diseases (IBDs). 

IBDs are characterized by severe chronic intestinal inflammation and imbalanced gut microbiota, or so-called dysbiosis. The etiology of these diseases is still unknown, but the current paradigm is that genetic and environmental factors, microbiome dysbiosis, and exacerbated immune responses against the gut microbiota act together to promote their development [8]. Alarming data indicate that the prevalence of IBDs is rising and impacting the quality of life of patients while being an economic burden for society [9]. With the approach of the personalized medicine era and being given new therapeutic options, it appears adequate to treat patients acknowledging their sex. Although sex-specific characteristics in IBDs are scarce and conflicting, some studies have reported differences in disease onset, associated with a more frequently found single-nucleotide polymorphism in ATG16L1 in females [10], development, and therapeutic sensitivity, where males have been reported to have response loss to anti-TNF drugs while females seem to develop more side effects [11]. IL10 promoter polymorphisms in women are associated with lower production of this anti-inflammatory cytokine and with increased susceptibility to IBDs at a younger age [12]. 

Besides the clear impact on immune responses, evidence has suggested sex as an important factor to consider when looking at gut microbiome interactions with the environment and the host. Among others, Org et al. demonstrated sex hormones to partially mediate the differences found in gut microbiota composition between females and males [13]. 

Despite the extensive utilization of IL-10^−/−^ mice as a model of experimental colitis in research, there is a need to further investigate and explore the sex differences in the IL-10 anti-inflammatory pathway response and their impact on the microbiota. Currently, the sex differences in the phenotype of IL-10^−/−^ mice remain unexplored, which represents an important gap in our understanding of this IBD model and its implications for colitis research. Here, we originally identified fundamental sex differences in the colitis phenotype of IL-10^−/−^ mice. Female mice exhibited a higher susceptibility to developing intestinal inflammation compared to males, and this was accompanied by distinct differences in their microbiota composition. Specifically, the microbiota of female mice displayed characteristics commonly associated with a detrimental phenotype, including a higher abundance of the phylum Bacteroidetes and a lower abundance of Firmicutes. Our findings highlight that sex might be chosen wisely in animal experimentation by considering sex specificities. 

## 2. Results

### 2.1. Female IL-10^−/−^ Mice Develop a More Severe Colitis Phenotype Than Males 

The central hypothesis this study sought to test was whether sex plays a role in the phenotype of colitis developed in the murine model of IL-10^−/−^ mice. We used IL-10^−/−^ mice, known to spontaneously develop chronic enterocolitis with massive infiltration of lymphocytes, activated macrophages, and neutrophils [7]. A total of 29 IL-10^−/−^ mice (14 males and 15 females) were monitored from baseline (4 weeks old) for 13 weeks (Figure 1A). Once a week, feces were collected, body weights were registered (Figure 1B), and survival rates were recorded (Figure 1C). As expected, males gained more body weight than females (mean: ♂ 14.9 g vs. ♀ 10.7 g). However, females had a slightly lower probability of survival (80%, final n = 12) than males (92.8%, final n = 13) during the first 17 weeks of life. Upon a general inspection when collecting the gastrointestinal organs, more severe inflammation was macroscopically observed in females. The colon length (Figure 1D) was significantly reduced while the ratio of the colon weight/length (Figure 1E) was slightly increased in females. The spleen weight/body weight ratio (Figure 1F), widely used as a marker of systemic inflammation in mice and recently demonstrated associated with disease activity in IBD patients [14], was significantly increased in female IL-10^−/−^ mice. 

The measurement of fecal lipocalin-2, known as a dynamic marker of inflammation mainly secreted by neutrophils, was used as a non-invasive method to confirm the development of colitis in all the mice over time (Figure 2A). Genetically modified mice exhibit the onset of intestinal inflammation at 8 weeks of age, which intensified and reached its maximum sustained activity by the time they were 12 weeks old. IL-10^−/−^ mice were euthanized at 17 weeks of age and characterized according to the amounts of their pro-inflammatory markers. As shown in Figure 2B–E, intestinal inflammation was more severe in female mice. The sex-based differences in intestinal inflammation were confirmed by the measurement of MPO (Figure 2B), a pro-inflammatory marker of granulocytes infiltration mainly expressed in neutrophils. The analysis of cytokines by RT-qPCR showed an increased production of TNF (Figure 2C), IL1β (Figure 2D), IL6 (Figure 2E), and TGFβ (Figure 2F) in female IL-10^−/−^ mice compared to IL-10^−/−^ males. 

Although no microscopic differences were found between colonic tissues after H&E staining and histologic examination (Appendix A), our findings together indicate the presence of inflammation with a higher grade in female IL-10^−/−^ compared to males. 

### 2.2. IL-10^−/−^ Mice Colitis Undergoes Sex-Specific Microbiota Changes

Since we observed a more aggressive inflammatory phenotype among females, and given the previous reports of both inflammation [15,16] and sex dependency on microbiota composition and function [13,17], we assessed how the microbiota was affected in each sex during the development of colitis. First, we measured the fecal amounts of LPS and flagellin during the study. LPS are the major components of the outer membrane of gram-negative bacteria, while flagellin is the structural component of flagella. These bacterial components are commonly increased in colitis-associated microbiota and linked with enriched levels of motile bacteria [15,18]. Interestingly, fecal amounts of both LPS and flagellin (Figure 3A) had an opposite evolution, significantly increasing in males and decreasing in females, illustrating sex-dependent discrepancies in the functional parameters of the microbiota. Further analysis of the microbiota after 16S rRNA sequencing followed by PCoA of the Bray–Curtis matrix of male IL-10^−/−^ mice revealed progressive change in the microbiota, evolving (Figure 3B, left graph). Contrastively, female IL-10^−/−^ mice had very pronounced microbiota shifts that could be clearly clustered before colitis (D0) and during mild colitis (D42) and severe colitis (D91, Figure 3B, right graph). The analysis of distances between samples within or between time groups also showed that microbiota composition in females was significantly more impacted during severe colitis (D91 vs. D0) than in males (Figure 3C). 

Next, the taxonomic analysis revealed that the Firmicutes/Bacteroidetes phylum ratio, widely accepted to have an important influence on maintaining normal intestinal homeostasis, was significantly reduced in female IL-10^−/−^ mice during mild and severe colitis and significantly lower than that of males at D42 and D91 (Figure 3D). The genus *Oscillospira* was the main contributor to the observed decrease in *Firmicutes* (Figure 3E, left), and *Rikenellaceae* was the family with most weight that contributed to the increase in *Bacteroidetes* (Figure 3E, right). Remarkably, the respective decrease and increase in these bacteria were significantly more pronounced in female IL-10^−/−^ mice than in males. 

Furthermore, we analyzed the relative abundance of the microbiota at the order level and found a clear increase in *Verrucomicrobiales* (Figure 3F) in female IL-10^−/−^ mice over time, which corresponded with the very significant increase in the specie *Akkermansia muciniphila*. The full taxonomic panel of relative abundances at the specie level can be found in Appendix A. Our observation suggests that microbiota could potentially drive sex differences in the colitic phenotype of IL-10^−/−^ mice. 

### 2.3. Expression of Fecal miR-21 Is Influenced by Sex

The latest research in IBDs has explored miRNAs as new factors that may be playing an important role in regulating both the host and the microbiota [19]. Our group has previously identified that the fecal miRNA signature is associated with the inflammatory potential of the microbiota in IL-10^−/−^ mice [16]. Among the identified altered miRNAs, miR-21 was found to be significantly altered in the presence of colitis [20]. Thus, we measured the fecal levels of this miRNA before and during severe colitis and observed an increase in fecal miR-21, but only in female IL-10^−/−^ mice (Figure 4A). More interestingly, fecal levels of miR-21 served as a predictor marker of severe inflammation (Figure 4B). Although no sex differences of this miRNA were found in colonic tissues (Figure 4C), miR-21 amounts positively correlated with macroscopic colonic inflammation in female IL-10^−/−^ mice (Figure 4D), supporting the idea of a possible sexual dimorphism in miRNAs action. Given our previous report that fecal miRNAs, including miR-21, do correlate with specific microbiota members [16], our data suggest that both actors could be driving sex differences in the appearance of colitic phenotype of IL-10^−/−^ mice. 

## 3. Discussion

Sexual dimorphism is a prevalent feature observed in numerous common disorders such as cardiovascular diseases [21], osteoporosis [22], and autoimmune diseases [23]. It is well known that females exhibit stronger immune responses compared with males [2]. Although the potential role of sex as a modulatory factor in the development and progression of IBDs has been increasingly acknowledged in recent years, the available data remain inconclusive. The central hypothesis this study sought to test was whether sex plays a role in the phenotype of colitis developed in a murine model of IBDs, since proper documentation characterizing the sex-based differences in IL-10^−/−^ mice is lacking. 

The surveillance of female and male IL-10^−/−^ mice up to 17 weeks of age demonstrated that female IL-10^−/−^ are more prone to severe colitis and dysbiosis. Thus, this study demonstrated that IL10 deficiency drives sex-specific immune and microbial changes during intestinal inflammation development (Appendix A). These findings provide novel and important insight into sex-based differences in the pathophysiology of colitis which should be considered for in vivo experimental designs. To our knowledge, we have provided the first characterization of sex-based differences in the development of colitis in IL-10^−/−^ mice.

The first generation of the engineered IL10 deficient mice was performed through the mutation of the Il10^tm^ [1]^Cgn^ by Dr. Werner Muller [7]. In 1993, this group characterized the IL-10^−/−^ generated mice and reported, without distinguishing females from males, retarded growth, anemia, and enterocolitis with extensive mucosal hyperplasia, inflammatory reactions, as well as aberrant expression of major histocompatibility complex class II molecules on epithelia [7]. Since then, these IL10 knock-out mice have been the most used murine experimental models of IBDs and widely studied in several immunopathologies. In order to confirm disease states, we collected and studied the gastrointestinal organs of IL-10^−/−^ mice. Both, macroscopic (Figure 1D–F) and molecular markers (Figure 2A–F) were measured and revealed consistent and more severe inflammation in females compared to males at 17 weeks of age. Supporting our results, Tso et al. demonstrated sex-dependent metabolic profiles in IL-10^−/−^ mice and hypothesized these to be involved in the development and severity of intestinal inflammation. Indeed, authors reported more severe intestinal inflammation in female mice compared to males, but only by histological injury score and colonic interferon-gamma (IFNγ) secretion. Similar sexual dimorphism was described in the less used IBD model 129 Rag2^−/−^ mice [24]. In this scenario, however, the concomitant infection of *Helicobacter pylori* enhanced the colonic inflammatory response in males, resulting in more severe colitis and dysplasia compared with females [24]. We must remark that the fecal lipocalin-2 levels (Figure 2A), used as a non-invasive inflammatory marker, did not show significant differences between males and females. As Gondim Prata et al. previously demonstrated, lipocalin-2 may reflect enterocyte damage as well as neutrophil presence [25].

Part of the phenotypical variations found in the susceptibility and severity of colitis could be explained by immunological sex differences. Gunasekera et al. recently showed that the development of colitis in IL-10^−/−^ mice is dependent on IL-22 [26], a cytokine known to promote the restoration of harmed tissues by stimulating the regeneration of the intestinal epithelium [27]. Together with IL-17A, these two molecules regulate the body’s natural antimicrobial defenses to protect against enteric pathogens [28]. Although Gunasekera et al.’s work was based only on IL-10^−/−^ male mice between 16 to 20 weeks of age, studies have reported a significantly enhanced production of IL-22 in the liver of females compared to male mice [29]. Interestingly, testosterone seems to regulate IL-22 production in the female liver [29], playing sex-dependent immunosuppressive effects [30]. Estrogens, on another hand, have a complex role in inflammation and autoimmune disorders, with experimental evidence suggesting their involvement in gastrointestinal functions, potentially through estrogen receptor beta (ERβ). As shown by Goodman et al., female ERβ knockout mice treated with dextran sodium sulfate (DSS) exhibit a lower risk of intestinal inflammation when compared to male mice [31]. Furthermore, research involving humans has also indicated a connection between ERβ expression and the clinical manifestation of IBDs. Patients with active disease had lower expression of ERβ in peripheral immune cells compared to both patients in remission and healthy controls [32]. It is worth noting that this particular study did not assess differences based on age or sex [32]. 

Since the gut microbiota interacts with the host’s immune system, it is reasonable to assume that the sex differences in gut microbiota composition is associated with the disparities observed in immune responses. Indeed, studies have shown that early-life microbial exposures determine sex hormone levels, regulate autoimmune disease fate in individuals with high genetic risk, and modify progression to the autoimmunity commensal microbial community [17]. Moreover, the emergence of sex differences in gut microbiota during puberty suggests that sex hormones may contribute to shaping the composition of the gut microbiota. The removal of the androgen source by castration results in a gut microbiota profile resembling that of female mice rather than male mice [33]. Herein, we found that females, who exhibited a more aggressive inflammatory phenotype, had an opposite trend of fecal LPS and flagellin amounts compared to males (Figure 3A). Microbiota analysis revealed distinct changes in composition and function between the sexes (Figure 3B–G). Females experienced more pronounced shifts in microbiota composition during severe colitis compared to males. The Firmicutes/Bacteroidetes ratio (Figure 3D), important for intestinal homeostasis, was reduced in females during both mild and severe colitis, mainly due to the decrease in Firmicutes *Oscillospira* and the increase in Bacteroidetes *Rikenellacea* (Figure 3E). Supporting our sex-dependent findings, Son et al. [34] showed an increased Firmicutes/Bacteroidetes ratio in male IL-10^−/−^ mice compared with females despite the fact that their provider was different (Orient Bio Inc., Seongnam, Republic of Korea vs. Jackson Laboratory, Bar Harbor, ME, USA). The authors also described an increase in the phylum Proteobacteria in female mice [34], which we did not detect. Differences could be explained by different inflammatory statuses; however, the authors did not report any inflammatory data. Another evident difference between studies is the animals’ age (8 [34] vs. 17 weeks of age). Interestingly, differences detected between the bacterial population of males and females measured at day 42 were not significantly different at day 91. Extrapolating from lipocalin-2 measurements, inflammation was not as exacerbated on day 42 as on day 91 (Figure 2A). This suggests that dysbiosis may occur prior to inflammation and in a more important manner in females. After 91 days, both female and male mice presented severe chronic inflammation, which could influence stabilizing microbiota. Moreover, female mice also had a significant increase in *Verrucomicrobiales*, particularly *Akkermansia muciniphila*. These findings suggest that microbiota may play a role in driving the sex differences observed in the colitic phenotype of IL-10^−/−^. Of note, *Akkermansia muciniphila* has been described to play important anti-inflammatory roles by modulating metabolic pathways in diabetes, obesity, and others [35,36,37]. Nevertheless, its complex role is still being scrutinized. For instance, constipation-predominant irritable bowel syndrome patients (C-IBS) present an increase in the relative abundance of *Akkermansia muciniphila*, like our female mice, with surprisingly anti-inflammatory effects [38]. In accordance with our research, Seregin et al. [39] reported that *Akkermansia muciniphila* can induce intestinal inflammation in IL-10^−/−^ mice. The authors attributed this phenomenon to the disruption of the host’s immune tolerance towards the normal microbial community in the context of the disease, transforming *Akkermansia muciniphila* from a commensal microorganism to a pathobiont that facilitates colitis development in genetically vulnerable hosts [39].

Studies involving probiotics have demonstrated distinct inflammatory responses between female and male mice. For instance, the administration of *Lactobacillus farciminis* to female rats subjected to stress resulted in a significant reduction in the colonic mucosal mast cell count and inflammatory cytokine levels [40]. Similarly, sex differences in response to the administration of probiotic *Lactobacillus animalis NP-51* were observed in terms of cytokine responses, intestinal metabolic profiles, and gut microbiota in *Mycobacterium*-infected mice [41]. Nevertheless, research has indicated that the influence of sex on gut microbiota composition [42] might be contingent upon the genotype of the host [13]. 

Currently, the involvement of miRNAs in mediating sex disparities in diseases has received limited attention remaining understudied [43], and the conventional analysis fails to acknowledge the presence of dysregulated miRNAs specific to sex. Nonetheless, estrogens have been shown to regulate numerous miRNAs in various cellular contexts [44]. Additionally, the X chromosome contains a significant number of miRNAs that play crucial roles in diverse physiological processes, particularly immune function [45]. Thus, it is plausible that miRNAs contribute to sex-specific responses in IBD prevalence, progression, and outcome. To our knowledge, there are no previous reports examining the sex-distinct expression of fecal miR-21 in the context of IBDs. In our previous work, we identified a distinct fecal miRNA signature associated with the inflammatory potential of the microbiota in IL-10^−/−^ mice [20]. Among the miRNAs that showed altered expression, miR-21 was significantly affected in the presence of colitis [16]. Here, we measured the levels of fecal miR-21 before and during severe colitis and observed a significant increase specifically in female IL-10^−/−^ mice (Figure 4A). Notably, the levels of fecal miR-21 served as a predictive marker for severe inflammation (Figure 4B). While no sex differences were observed in miR-21 colonic expression (Figure 4C), there was a positive correlation between miR-21 levels and macroscopic colonic inflammation in female IL-10^−/−^ mice (Figure 4D). These findings support the notion of potential sexual dimorphism in the action of miRNAs. Corroborating our sex-bias findings, studies conducted in the field of cancer have demonstrated correlations between miRNAs and specific genders that are dependent on the stage of cancer: miR-24 in women, and miR-17 and miR-20 in men [46]. In recent research focused on long-term renal function in kidney transplant patients, it was found that female patients exhibited significantly worse outcomes, which were associated with distinct miRNA profiles [47]. A recent study has evaluated fecal miRNA expression levels in relation to sex, among other factors, and identified nine miRNAs whose expression in stool was different between sex [48]. Together with our present findings, these suggest the importance of sex as a possible confounder in miRNA-based studies in mice and humans. 

Overall, our study in a mouse model of colitis has the potential to significantly contribute to the understanding of sex differences in colitis development in mice and IBD patients. The purpose of rodent systems is to be experimentally more manageable than human subjects, allowing for the establishment of causal relationships between variables, such as sex, to be proven. However, despite the conserved underlying biological mechanisms to explain sex differences in animals and humans, it should not be assumed a priori that findings from mice will directly translate to the human condition. Thus, newly identified concepts about sex-driven aggravated conditions arising from animal experiments need to be validated in humans. 

## 4. Material and Methods

### 4.1. Mice Experiment and Housing 

Interleukin-10 knock-out or IL-10 KO (IL-10^−/−^) mice were originally purchased from Jackson laboratory (B6.129P2-Il10^tm^ [1]^Cgn^/J, Strain #:002251) and bred in-house. A total of 29 in-house-bred IL-10^−/−^ 4-week-old mice, 14 males and 15 females, were kept until 17 weeks of age under SPF conditions with a controlled temperature (25 °C) and photoperiod (12:12 h light–dark cycle). Mice were randomly divided into groups of 4–6 per sex and had ad libitum access to food and water. The procedures were approved by our local Animal Ethics Committee and the French Ministry of Research in accordance with the European legislation (APAFIS#23855-2020012916328162, Paris, France). Mice were euthanized after 13 consecutive weeks of monitoring (day 91). The schematic experiment design is represented in Figure 1A. 

### 4.2. Sample Collection and Preparation 

Every 7 days, body weight and fresh fecal samples were collected for further analysis. Macroscopic evidence of inflammation was evaluated by measuring the colon length, colon weight, and spleen weight. Colonic samples were collected for further analysis, as detailed below. 

### 4.3. Preparation of Fecal Supernatant

Fecal samples were reconstituted in phosphate-buffered saline (PBS) to a final concentration of 100 mg/mL, vortexed for 15 min, and centrifuged for 10 min at 14,000× *g* 4 °C. Fecal supernatant was collected, serially diluted, and stored at −80 °C until further use.

### 4.4. Quantification of Fecal Lipocalin-2 by ELISA

Lipocalin-2 was measured in previously processed fecal supernatant following the manufacturer’s instructions of the murine Lipocalin-2/NGAL ELISA kit (R&D Systems, Minneapolis, MN, USA). Optical density was read at 450 nm (VersaMax microplate reader, Molecular Devices, San Jose, CA, USA). 

### 4.5. Colonic Myeloperoxidase Assay

The activity of myeloperoxidase (MPO), a neutrophil enzyme and marker of inflammation, was analyzed as previously described [49]. Distal colonic tissues were mechanically homogenized at 50 mg/mL in 0.5% hexadecyltrimethylammonium bromide (Sigma-Aldrich, Saint-Quentin-Fallavier, France) diluted in 50 mM PBS (pH 6.0), sonicated, freeze–thawed 3 times, and centrifuged for 15 min at 14,000 rpm 4 °C. The collected supernatant (50 μL) was mixed with freshly prepared reactive buffer (200 μL) consisting of 1 mg/mL of dianisidine dihydrochloride (Sigma) and 5 × 10^−4^% H_2_O_2_. The change in optical density was measured at 450 nm in a SPARK 10M plate reader (Tecan, Männedorf, Switzerland). Human neutrophil MPO (Sigma) was used to create the standard curve. 

### 4.6. Colonic mRNA Isolation and Quantitative Real-Time qPCR Assays 

Colonic tissues were collected during euthanasia, placed in RNA-later (Invitrogen, Waltham, MA, USA), and kept at −80 °C until further use. A NucleoSpin RNA kit was used to isolate the total mRNA from colonic samples according to the manufacturer’s instructions. Extracted mRNAs were quantified using a NanoDrop One Spectrophotometer (Ozyme, Saint-Cyr-l’École, France). After cDNA synthesis, inflammatory cytokines were measured by RT-qPCR with TaqMan using a LightCycler 96 system (Roche, Basel, Switzerland) using the following primers: B2m, as a housekeeping gene, id: Mm00437762_m1; TNF superfamily id: Mm00443258_m1; IL6 id: Mm00446190_m1; IL1β id: Mm00434228_m1. TGF-beta level was analyzed by PCR with SYBR Green (Qiagen, Les Ulis, France) with the following primers: TGFb-fw: ATGCTAAAGAGGTCACCCGC and TGFb-rv: TGCTTCCCGAATGTCTGACG. Changes in mRNA expression were determined by calculating the fold changes using the comparative threshold cycle (Ct) method. 

### 4.7. Fecal and Colonic miRNA Extraction and Quantification 

Small RNA molecules (<200 nt) were obtained from the colonic and fecal samples collected using the mirVana isolation kit (Thermo Fisher, Waltham, MA, USA) according to the manufacturer’s instructions. Briefly, samples were disrupted and homogenized in a lysis buffer using TissueLyser (Qiagen, Les Ulis, France). The homogenate was mixed with chloroform and centrifuged. The aqueous phase was mixed with 1/3rd volume of ethanol and loaded into an RNeasy spin column. The collected filtrate was mixed with 2/3rd volumes of ethanol and filtered for a second time. After 3 washes, small RNAs were eluted with RNase-free water. The quality and quantity of small RNAs were analyzed using a NanoDrop One Spectrophotometer (Ozyme, France). The miRCURY LNA RT kit (Qiagen) was used for the cDNA synthesis from the pre-diluted small RNA samples (10 ng/μL). The obtained cDNA was mixed with the 2× miRCURY SYBR Green Master Mix and qPCR was performed in a LightCycler 96 system (Roche). The UniSp6 RNA spike-in was used as an interpolate calibrator. Changes in miRNA expression were determined by calculating the fold changes using the comparative threshold cycle (Ct) method with a normalization with miR-194-5p, chosen for its stability index in colitic mice. 

### 4.8. Quantification of Fecal LPS and Flagellin Load

The fecal load of LPS and flagellin were quantified using HEK-Blue-mTLR4 and HEK- Blue-mTL5 cells, respectively (Invivogen, San Diego, CA, USA). The previously processed fecal supernatant was applied to the mammalian cells and incubated for 24 h at 37 °C. Cell culture supernatants were applied to QUANTI-Blue medium (Invivogen) and alkaline phosphatase activity was measured at 620 nm after 30 min. Purified LPS from *E. coli* (Sigma) and flagellin from *Salmonella typhimurium* (Sigma) were used for standard curve determination. 

### 4.9. Microbiota Analysis by 16S rRNA Gene Sequencing Using Illumina Technology 

Microbiota analyses were performed before developing colitis (day 0), during mild colitis (day 42), and during severe colitis (day 91). The 16S rRNA gene amplification and sequencing were performed using the Illumina MiSeq technology following the protocol of the Earth Microbiome Project with their modifications to the MOBIO PowerSoil DNA Isolation Kit procedure for extracting DNA (https://press.igsb.anl.gov/earthmicrobiome (accessed on 1 April 2023)). Bulk DNA was extracted from frozen feces using a PowerSoil-htp kit from MoBio Laboratories (Carlsbad, CA, USA) with mechanical disruption (bead-beating). The 16S rRNA genes, region V4, were RT-qPCR amplified from each sample using a composite forward primer and a reverse primer containing a unique 12-base barcode, designed using the Golay error-correcting scheme, which was used to tag RT-qPCR products from the respective samples). We used the forward primer 515F 5′-AATG ATAC GGCG ACCA CCGA GATC TACACGCT XXXX XXXX XXXX TATG GTAATT GT GTGYCAGCMGCCGCGGTAA-3′: the italicized sequence is the 5′ Illumina adaptor, the 12 X sequence is the Golay barcode, the bold sequence is the primer pad, the italicized and bold sequence is the primer linker, and the underlined sequence is the conserved bacterial primer 515F. The reverse primer 806R used was 5′-CAAG CAGA AGAC GGCA TACGAGAT AGTCAGCCAG CC GGACTACNVGGGTWTCTAAT-3′: the italicized sequence is the 3′ reverse complement sequence of Illumina adaptor, the bold sequence is the primer pad, the italicized and bold sequence is the primer linker, and the underlined sequence is the conserved bacterial primer 806R. RT-qPCR reactions consisted of Hot Master RT-qPCR mix (Quantabio, Beverly, MA, USA), 0.2 mM of each primer, 10–100 ng template, and reaction conditions were 3 min at 95 °C, followed by 30 cycles of 45 s at 95 °C, 60 s at 50 °C, and 90 s at 72 °C on a Biorad thermocycler. RT-qPCR products were purified with Ampure magnetic purification beads (Agencourt, Brea, CA, USA) and visualized by gel electrophoresis. Products were then quantified (BIOTEK Fluorescence Spectrophotometer, Winooski, VT, USA) using Quant-iT PicoGreen dsDNA assay. A master DNA pool was generated from the purified products in equimolar ratios. The pooled products were quantified using Quant-iT PicoGreen dsDNA assay and then sequenced using an Illumina MiSeq sequencer (paired-end reads, 2Å~250 bp, San Diego, CA, USA) at Cornell University, Ithaca. The 16S rRNA sequences were analyzed using QIIME2-version 2019.51 Sequences were demultiplexed and quality filtered using the DADA2 method with QIIME2 default parameters to detect and correct Illumina amplicon sequence data, and a table of Qiime2 artifacts was generated. A tree was next generated, using the align-to-tree-mafft-fasttree command, for phylogenetic diversity analyses, and alpha and beta diversity analyses were computed using the core-metrics-phylogenetic command. PCoA plots were used to assess the variation between the experimental group (beta diversity). 

### 4.10. Statistical Analysis

Data are expressed as mean ± SEM and statistical analyses were performed using GraphPad Prism software (V.8). Significance was determined using unpaired *t*-tests or the Mann–Whitney U test when data were normally or non-normally distributed, respectively. For data collected at different time points in line chart form, a two-way repeated-measures ANOVA or a mixed-effects model (if missing values) with a Bonferroni post hoc test was performed. The Spearman correlation coefficient matrixes were used to screen for the relationship between variables. Linear regressions of the most relevant identified conditions were represented and analyzed using linear regression. Significance was noted as follows: compared to day 0, * *p* < 0.05, ** *p* < 0.01, *** *p* < 0.001, and **** *p* < 0.0001. Males vs. females at each time point, # *p* < 0.05, ## *p* < 0.01, ### *p* < 0.001, #### *p* < 0.0001. Results were considered significant at *p* < 0.05. 

## 5. Conclusions

While sex plays a crucial role in disease, a substantial number of sex-specific analyses tend to concentrate on one sex rather than emphasizing the comparative aspect. As a result, the field of sex and gender medicine is often underexplored, and there is a growing body of literature highlighting the necessity of including both sex in animal models, clinical trials, and healthcare planning policies. Gaining detailed knowledge about sex-related microbial and immunoregulation remains a crucial prerequisite for the development of adequate disease models and therapeutic strategies enabling personalized medicine.

## Figures and Tables

**Figure 1 ijms-24-10364-f001:**
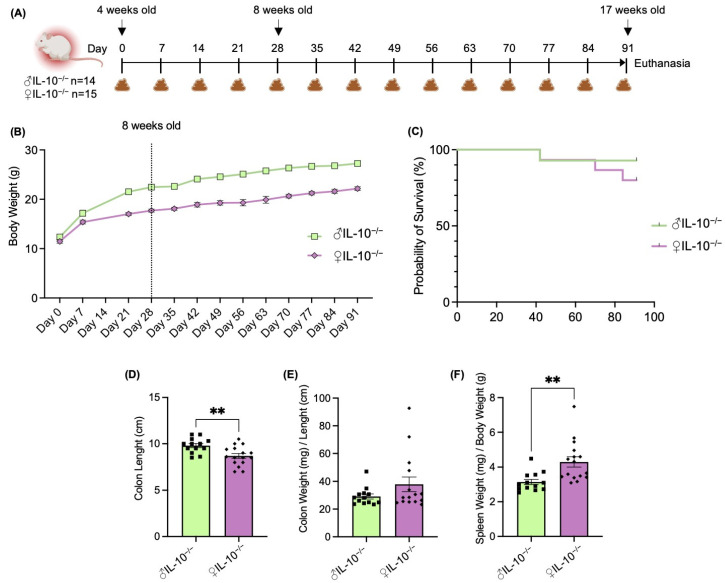
Female IL-10^−/−^ mice developed more severe colitis that was quantifiable macroscopically. (**A**) Schematic representation of the experimental design where the female (♀) and male (♂) interleukin 10 knockout (IL-10^−/−^) mice were monitored for 13 weeks. Feces and (**B**) body weight were collected every week, and (**C**) survival rate was calculated. At the endpoint, gastrointestinal organs were weighted and measured: (**D**) colon length, (**E**) colon weight/length ratio, and (**F**) spleen weight/body weight ratio. Data are represented as means ± SEM. Statistical analyses were performed using the Mann–Whitney U test. Significant differences were recorded as ** *p* < 0.01.

**Figure 2 ijms-24-10364-f002:**
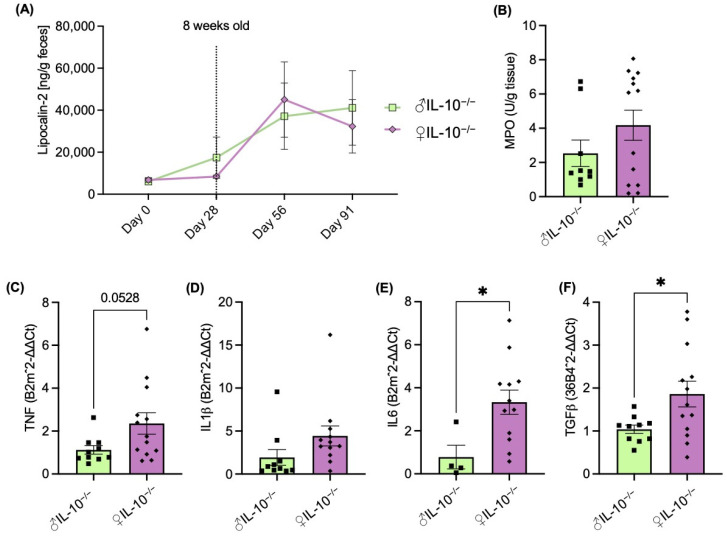
The molecular study of inflammatory markers showed female (♀) IL-10^−/−^ mice developed more aggressive colitis than males (♂). (**A**) Fecal lipocalin-2 level measured over time. The levels of (**B**) myeloperoxidase (MPO), (**C**–**F**) tumor necrosis factor (TNF), interleukin (IL)1β, IL6, and TGFβ were measured in colonic samples by colorimetric assay and RT-qPCR. Data are represented as means ± SEM. Statistical analyses were performed using the Mann–Whitney U test. Significant differences were recorded as * *p* < 0.05.

**Figure 3 ijms-24-10364-f003:**
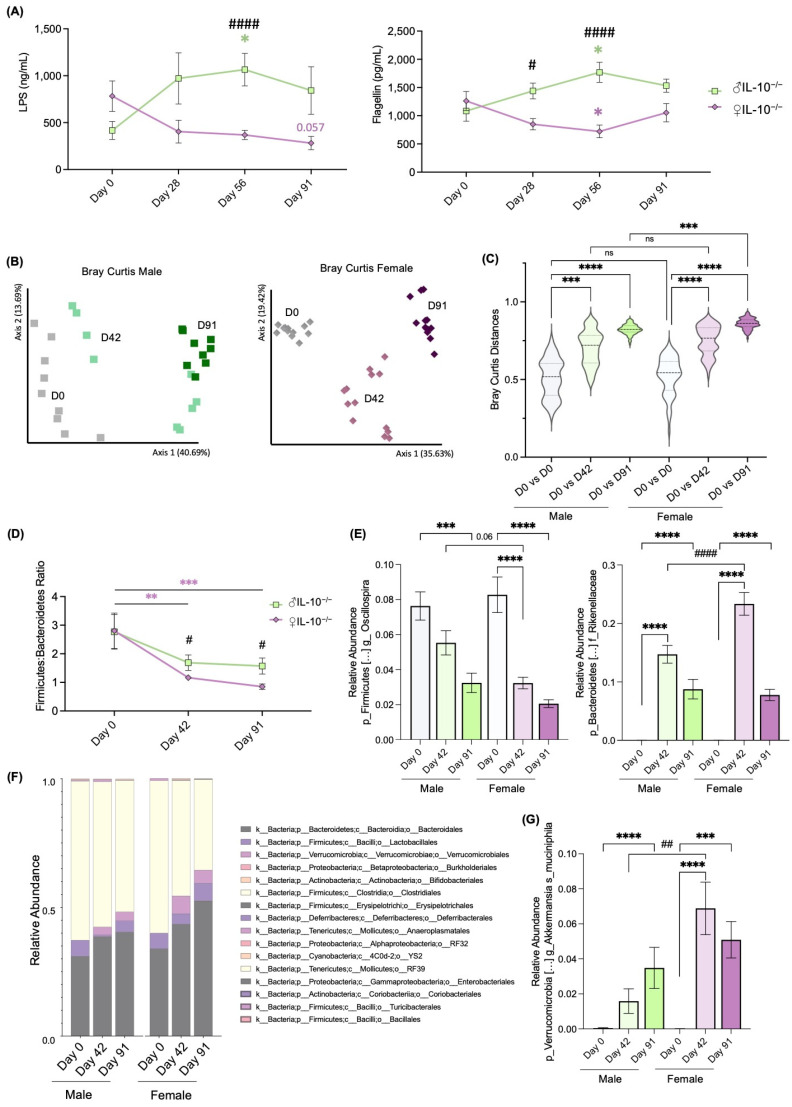
Dysbiosis associated with colitis was more aggravated in female IL-10^−/−^ mice compared to males. (**A**) Fecal lipopolysaccharide (LPS), measured using TLR4 HEK cells, and flagellin, measured using TLR5 HEK cells, diverged over time between females (♀) and males (♂). (**B**) Principal coordinates analysis (PCoA) and (**C**) analysis of distances of the Bray–Curtis matrix of males and females IL-10^−/−^ mice microbiota assessed by 16S rRNA gene sequencing at days 0, 42, and 91. (**D**) *Firmicutes/Bacteroidetes* ratio was more greatly modified in females, as a result of (**E**) the remarkable decrease in the genus *Oscillospira* and (**F**) the increase in the family *Rikenellaceae*. (**F**) Taxonomic summary at the order level, highlighting (**G**) the most significantly modified specie, *Akkermansia muciniphila*. For bar and linear graphs, values are given as mean ± SEM. Data represented in violin plots show median ± quartiles. Statistical analyses were performed using the two-way ANOVA test, followed by a Bonferroni post hoc test. Significant differences were recorded as * *p* < 0.05, ** *p* < 0.01, *** *p* < 0.001, **** *p* < 0.0001, when compared to day 0 (D0); and # *p* < 0.05, ## *p* < 0.01, #### *p* < 0.0001, when compared female vs. male at each time point.

**Figure 4 ijms-24-10364-f004:**
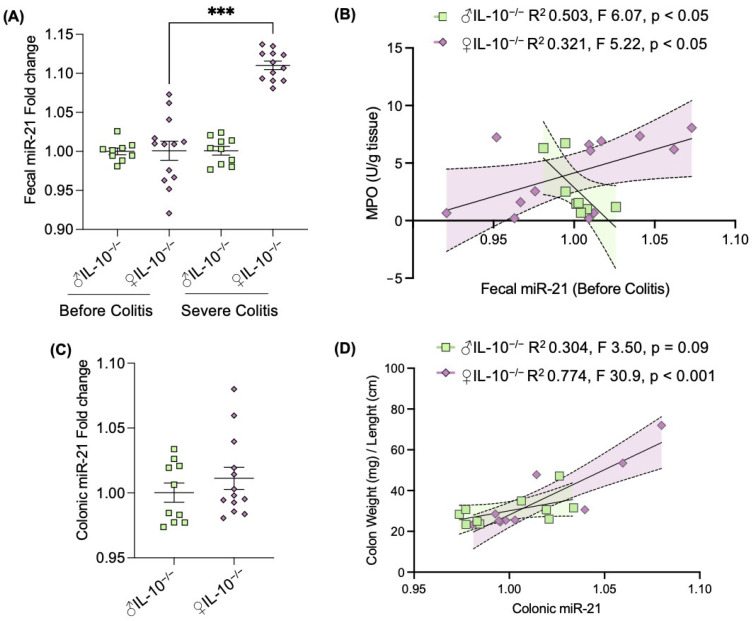
Expression of miR-21 was related to colonic inflammation in females (♀), but not IL-10^−/−^ males (♂). (**A**) Fecal miR-21 relative abundance was measured in stools from IL-10^−/−^ mice before and after developing colitis, from 5 and 17 weeks old, respectively. (**B**) Lineal regression between the two most significantly correlated variables, fecal miR-21 relative abundance before colitis and colonic myeloperoxidase (MPO) amounts during colitis demonstrated a significant relationship between variables, opposite in females compared to males. (**C**) Colonic relative abundance of miR-21 during colitis correlated with (**D**) the colon weight/length ratio. Dot plots are given as mean ± SEM. Statistical analyses were performed using the Mann–Whitney U test. Significant differences were recorded as *** *p* < 0.001. F-tests were used to determine the significance of the correlations. The differences in slopes and interactions between females and males were tested and shown as significantly different.

## Data Availability

Unprocessed sequencing data are deposited in the European Nucleotide Archive under accession number PRJEB63263.

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
