# Peer review of "The Effect of Sex-Specific Differences on IL-10−/− Mouse Colitis Phenotype and Microbiota"

_ijms, 2023, doi:10.3390/ijms241210364_

Round 1
Reviewer 1 Report
I have reviewed in detail the paper entitled: “Sex-specific difference on IL10-/- mice colitis phenotype and microbiota”. This is an original article in which the authors evaluated the differences between the degree of colitis and the changes in the microbiota in an IL-10 KO mouse model-comparing males vs. females. I found it an interesting article; however, I have some comments:
1. Did any mice die during the study? Or did they all survive 17 weeks?
2. I would have liked to see the survival graph, at least in the supplements section
3. In Figure 2, they mention the following: “The molecular study of inflammatory markers confirmed female (♀) IL-10-/- mice developed more aggressive colitis than males (♂)”. However, they only showed significant differences in one inflammatory marker (IL-6). Why only differences in IL-6?
4. It has been described that lipocalin-2 is related to the severity of the disease, however, they showed an increase in this marker over the days, but no apparent difference between males vs. females.
5. Why evaluate the cytokines TNF-alpha, IL-1, and IL-6 and not others such as those related to the Th17 profile described in autoimmune processes of the gastrointestinal tract?
6. Why evaluate the mRNA and not the protein? I think that the determination of mRNA is usually relative because not all mRNA is converted into protein, which is the active part. Perhaps an immunohistochemical assay would have been appropriate.
7. In Figure 3, the authors mention that the Firmicutes/Bacteroidetes ratio decreases and they are correct, however, no significant differences are observed between males vs. females, or at least they did not place it in the figure (Figure 3D).
8. They showed differences between the bacterial populations of males vs. females at 42 days, but not at 91 days, why? This was not mentioned in the discussion (Figure 3E)
9. Why didn't you search for a differential response in anti-inflammatory cytokines like TGF-beta that could support the idea of less severe colitis in males vs. females?
10. These results are different from those published by Son HJ et al., where they demonstrated that the Firmicutes/Bacteroidetes ratio was higher in the group of males vs. females.
11. They also mentioned increased phylum Proteobacteria in female IL-10 KO mice. I consider including that part in the discussion (Reference: Son HJ, Kim N, Song C, Nam RH, Choi SI, Kim JS, Lee DH. Sex-related Alterations of Gut Microbiota in the C57BL/6 Mouse Model of Inflammatory Bowel Disease. J Cancer Prev 2019;24:173-182. https://doi.org/10.15430/JCP.2019.24.3.173)
Minor editing of the English language required
Author Response
Please find a revised version of our manuscript entitled “Sex-specific difference on IL-10-/- mice colitis phenotype and microbiota”. Our initial submission was very well received by the two reviewers. Following the reviewers’ recommendations, we have revised our manuscript. Please see below for our point-by-point response to the reviewers’ comments. Our changes are tracked in the revised manuscript.
Reviewer 1:
I have reviewed in detail the paper entitled: “Sex-specific difference on IL10-/- mice colitis phenotype and microbiota”. This is an original article in which the authors evaluated the differences between the degree of colitis and the changes in the microbiota in an IL-10 KO mouse model-comparing males vs. females. I found it an interesting article; however, I have some comments:
- Did any mice die during the study? Or did they all survive 17 weeks?
Some mice did not survive the total experiment. A survival curve has now been included as Figure 1C. A total of one male and three females did not survive across the 17 weeks of surveillance. The survival rate was 92.8% for males and 80% for females (although no significant difference was detected).
- I would have liked to see the survival graph, at least in the supplements section
As mentioned in the answer to the previous comment, a survival curve in now included as main Figure (1C). We thank the reviewer for this recommendation.
- In Figure 2, they mention the following: “The molecular study of inflammatory markers confirmed female (♀) IL-10-/- mice developed more aggressive colitis than males (♂)”. However, they only showed significant differences in one inflammatory marker (IL-6). Why only differences in IL-6?
Title of figure 2 was changed for more accuracy: ‘Figure 2. The molecular study of inflammatory markers showed female (♀) IL-10-/- mice developed a more aggressive colitis than males (♂)’. Moreover, we would like to highlight that levels of other (macroscopic and molecular) markers are also different (significantly or close to significance) in females compared to males.
- It has been described that lipocalin-2 is related to the severity of the disease, however, they showed an increase in this marker over the days, but no apparent difference between males vs. females.
In our study the use of fecal lipocalin-2 was the aimed to assess intestinal inflammation in a non-invasive way in the same manner as fecal Calprotectin in human. Although MPO and lipocalin-2 are both considered neutrophil markers, Gondim Prata et al showed MPO and Lcn-2 do not exactly correlate (doi: 10.15761/JTS.1000130). Authors justify it saying Lcn-2 may also reflect enterocyte damage as well as neutrophil presence. This has been added in the discussion (Lines 302-305).
- Why evaluate the cytokines TNF-alpha, IL-1, and IL-6 and not others such as those related to the Th17 profile described in autoimmune processes of the gastrointestinal tract?
Our main objective with this study was to identify general differences in inflammation between male and female rather than characterizing the inflammation. However, we have now included TGF-Beta in Figure 2F, as a central regulatory cytokine.
- Why evaluate the mRNA and not the protein? I think that the determination of mRNA is usually relative because not all mRNA is converted into protein, which is the active part. Perhaps an immunohistochemical assay would have been appropriate.
As differences were already observed at mRNA level, we then concluded that inflammation was differently affected between male and female and that we had achieved our overall objective of reporting differences male/female in IL10-/- mice. We however, thank the reviewer for the suggestion, it would be indeed an important question to address in a further study.
- In Figure 3, the authors mention that the Firmicutes/Bacteroidetes ratio decreases and they are correct, however, no significant differences are observed between males vs. females, or at least they did not place it in the figure (Figure 3D).
We apologize for omitting to specify this important difference. By comparing the ratio F/B at D91 (when mice had developed colitis), but also earlier at D42, we observed that it was significantly lower in female compared to mice. Those statistical differences were shown by a # in the figure and were mentioned in the results section (line 226). Importantly, the significance of such differential ratio is now discussed (line 341-359).
- They showed differences between the bacterial populations of males vs. females at 42 days, but not at 91 days, why? This was not mentioned in the discussion (Figure 3E).
Albeit lower inflammation in female than male, male still develop colitis and have alteration of their microbiota (per seen with LCN-2 levels). Comparison of the dynamics of the evolution of LCN-2 and LPS/FLIC (figure 3A and B), it seems that inflammation is minimal at D28 in both males and females despite already existing alteration of the microbiota function at D28. Together this suggests that dysbiosis might occur prior to inflammation and that dysbiosis, which we think might develop earlier in female than mice, have then reached stability and remain stable in both male and female. This important point is now extensively discussed ((line 341-359).
- Why didn't you search for a differential response in anti-inflammatory cytokines like TGF-beta that could support the idea of less severe colitis in males vs. females?
TGF-beta has been analyzed by qPCR and is now added on Figure 2F. We observed a higher level of this cytokine in females. This could be seen as a mean by which intestine of IL-10-/- females would dampen the otherwise exacerbated inflammation and regenerate the tissue per its described role on such phenomena.
- These results are different from those published by Son HJ et al., where they demonstrated that the Firmicutes/Bacteroidetes ratio was higher in the group of males vs. females.
Thank you for the observation. Indeed, Son et al reported a higher Firmicutes/Bacteroidetes ratio in males, just like us, supporting our results. This concordance with Son’s study, despite use of different mouse provider, was added in discussion, lines 343-351.
- They also mentioned increased phylum Proteobacteria in female IL-10 KO mice. I consider including that part in the discussion (Reference:Son HJ, Kim N, Song C, Nam RH, Choi SI, Kim JS, Lee DH. Sex-related Alterations of Gut Microbiota in the C57BL/6 Mouse Model of Inflammatory Bowel Disease. J Cancer Prev 2019;24:173-182. https://doi.org/10.15430/JCP.2019.24.3.173)
We would like to thank the reviewer for suggesting such a relevant publication. We have added it and discuss the main result differences, lines 352-3
Reviewer 2 Report
Maite et al. explored the sex differences in inflammatory phenotype and gut microbiome of IL-10-/- mice and found that female mice were more prone to develop intestinal inflammation than male mice. The topic of this paper is very interesting and some important results were revealed. The manuscript is well-written, but there are a few points that need to be clarified. Here are some comments on this paper and questions that I would like to discuss with the authors:
1. Lines 59-61 for sex-specific characteristics in IBD, could authors provide more detailed information? It will help readers to understand the differences between females and males with IBD.
2. Line 86 “A total of 29 IL-10-/- mice (14 males and 15 females) were monitored from baseline (4 weeks old) for 13 weeks (Figure 1A)” and line 90 “females had a slightly lower probability of survival (78.5%) than males (92.8%) during the first 17 weeks of life”. The number of mice used in the study is confusing, if 14 males and 15 females survived, the initial number of mice would be 14/0.92=15.2 males and 15/0.785=19.1 females.
3. Histologic examination is a method to see colon injury. Although there is no significant difference between male and female mice, it is advisable to provide H&E figures in the supplemental file.
4. Lines 271-272, “Female mice also showed a significant increase in Verrucomicrobiales, particularly Akkermansia muciniphila.” Many studies have shown Akkermansia muciniphila to be a probiotic with therapeutic effects for IBD. Could the authors give more discussions about Akkermansia enriched in female mice?
5. Line 352 the primer information for qPCR is missing.
6. There are some spelling mistakes, for example, line 115 PCR should be RT-qPCR, line 116 TNF should be TNF-α, line 337 14.000 g should be 14,000, line 344 underline of MPO, line 413 Dada2 should be DADA2 and needs to be referenced.
7. The clarity and resolution of the figures in the manuscript need to be improved, the figures in the supplementary file are quite clear and high-resolution.
Author Response
Maite et al. explored the sex differences in inflammatory phenotype and gut microbiome of IL-10-/- mice and found that female mice were more prone to develop intestinal inflammation than male mice. The topic of this paper is very interesting and some important results were revealed. The manuscript is well-written, but there are a few points that need to be clarified. Here are some comments on this paper and questions that I would like to discuss with the authors:
- Lines 59-61 for sex-specific characteristics in IBD, could authors provide more detailed information? It will help readers to understand the differences between females and males with IBD.
We would like to thank the reviewer for the suggestion. We have now added in the introduction some details about the main findings for the two articles (line 61-64).
- Line 86 “A total of 29 IL-10-/-mice (14 males and 15 females) were monitored from baseline (4 weeks old) for 13 weeks (Figure 1A)” and line 90 “females had a slightly lower probability of survival (78.5%) than males (92.8%) during the first 17 weeks of life”. The number of mice used in the study is confusing, if 14 males and 15 females survived, the initial number of mice would be 14/0.92=15.2 males and 15/0.785=19.1 females.
We apologize for the confusion. The number of mice at the beginning of the follow-up was 14 males and 15 females. Along the course of the surveillance, 1 male and 3 females have died, as now displayed in the newly added survival curve (Figure 1C). The number of mice at this end of the experiment was therefore 13 males and 12 females. The survival rate of female is actually 80%. It is now made clear in the manuscript (line 93-94 and Figure 1C).
- Histologic examination is a method to see colon injury. Although there is no significant difference between male and female mice, it is advisable to provide H&E figures in the supplemental file.
Histological scoring and representative H&E images were added as Supplementary Figure S1. We thank the reviewer for the suggestion.
- Lines 271-272, “Female mice also showed a significant increase in Verrucomicrobiales, particularly Akkermansia muciniphila.” Many studies have shown Akkermansia muciniphilato be a probiotic with therapeutic effects for IBD. Could the authors give more discussions about Akkermansia enriched in female mice?
Indeed, many studies have shown Akkermansia as a probiotic, especially in the context of human metabolic syndrome. Nevertheless, discrepancies still exist in the literature. We have now extensively discussed this matter in the context of IL-10 deficiency in mice (Lines 363-374).
- Line 352 the primer information for qPCR is missing.
We apologize for omitting this important information. The assay identities from TaqMan of each cytokine as well as the primers for TGF-beta, are now added in the corresponding method section (lines 466-469).
- There are some spelling mistakes, for example, line 115 PCR should be RT-qPCR, line 116 TNF should be TNF-α, line 337 14.000 g should be 14,000, line 344 underline of MPO, line 413 Dada2 should be DADA2 and needs to be referenced.
Thanks to the reviewer for pointing the spelling mistakes. We have corrected them all except for TNF. Although TNF has historically been referred to as “TNF-alpha” to distinguish it from “TNF-beta”, “TNF-beta” was renamed to lymphotoxin-alpha (LTα), and thus the distinction between alpha and beta is no longer necessary (PMID: 24636534 & PMID: 36358688).
- The clarity and resolution of the figures in the manuscript need to be improved, the figures in the supplementary file are quite clear and high-resolution.
We have now provided high quality figures.